# `SpecForge`: A Flexible and Efficient Open-Source Training Framework for Speculative Decoding

**Shenggui Li** [1][2]  **Chao Wang** [2][3]  **Yikai Zhu** [2]  **Yubo Wang** [2]  **Fan Yin** [2]  **Shuai Shi** [2]  **Yefei Chen** [2]
**Xiaomin Dong** [4]  **Qiaoling Chen** [1]  **Jin Pan** [2]  **Ji Li** [2]  **Yineng Zhang** [2]  **Lei Yu** [3]
**Yonggang Wen** [1]  **Ivor Tsang** [5]  **Tianwei Zhang** [1]

 GitHub: https://github.com/sgl-project/SpecForge

## Abstract

Speculative decoding mitigates the memory-bound nature of LLM decoding by using a lightweight draft model to propose multiple tokens for parallel verification. However, its adoption has been limited by the lack of high-quality draft models and scalable training infrastructure. We introduce `SpecForge`, an open-source and efficient framework for training speculative decoding models with full support for EAGLE-3. `SpecForge` incorporates target–draft decoupling, hybrid parallelism, optimized training kernels, and tight integration with production-grade inference engines, enabling up to $9.9\times$ faster EAGLE-3 training for Qwen3-235B-A22B compared to the baseline. We further release `SpecBundle`, a suite of production-grade EAGLE-3 draft models trained with `SpecForge` for mainstream open-source LLMs, achieving up to $4.48\times$ end-to-end inference speedup on SGLang and addressing the scarcity of high-quality drafts. Finally, we distill a systematic study of speculative decoding training into practical and actionable recipes to guide real-world adoption.

## 1. Introduction

Large language models (LLMs) have become a cornerstone of modern AI systems. Both proprietary models—such as ChatGPT (Achiam et al., 2023), Gemini (Reid et al., 2024; Comanici et al., 2025), and Grok—and open-source counterparts including LLaMA (Touvron et al., 2023a;b; Dubey et al., 2023), DeepSeek (DeepSeek-AI et al., 2024; 2025), and Qwen (Bai et al., 2023; Qwen et al., 2025; Yang et al., 2025) have driven substantial productivity gains across industries. However, as model sizes continue to scale, inference latency has emerged as a fundamental bottleneck (Yu & Jeong, 2022; Recasens et al., 2025). Autoregressive generation requires a full forward pass through billions of parameters for each token, yielding a memory-bound inference process that significantly increases deployment cost and limits real-time or high-throughput applications.

Speculative decoding accelerates LLM inference by pairing a lightweight draft model with a large target model, where the draft proposes multiple candidate tokens that are verified in parallel by the target model in a single forward pass (Leviathan et al., 2022; Chen et al., 2023). Draft models span a wide design space, including N-gram predictors (Fu et al., 2024), smaller models from the same family (Leviathan et al., 2022; Chen et al., 2023), sub-layers of the target model (Zhang et al., 2024a; Liu et al., 2024; Xia et al., 2024), and autoregressive adapters (Cai et al., 2024; Li et al., 2024b;a; Zhang et al., 2024b; Du et al., 2024; Li et al., 2025). Once accepted, draft tokens are committed in batches, reducing forward passes of the target model and exploiting parallelism to speed up the inference without altering output distributions.

Notably, EAGLE-3 (Li et al., 2025) leverages hybrid token features, dynamic candidate tree and Training-Time Test (TTT) to significantly increase token acceptance rates, achieving up to 4.79× speedup on LLaMA-3.3-70B without quality degradation. It should be noted that the speedups reported above are obtained using pure PyTorch/JAX implementations, without relying on mature production inference engines. Owing to its strong empirical performance, EAGLE-3 has become the de facto industrial standard for speculative decoding and is supported by major inference engines, including SGLang (Zheng et al., 2023), vLLM (Kwon et al., 2023), and TensorRT-LLM (trt, 2025).

Despite these advances, speculative decoding—particularly

[1]Nanyang Technological University  [2]SpecForge Team  [3]Meituan  [4]EigenAI  [5]A*STAR, Singapore. Correspondence to: Tianwei Zhang <tianwei.zhang@ntu.edu.sg>.

*Proceedings of the $43^{rd}$ International Conference on Machine Learning*, Seoul, South Korea. PMLR 306, 2026. Copyright 2026 by the author(s).

EAGLE-3—remains underutilized in practice. We identify three key factors limiting its broader adoption.

**Cause 1: Limited availability of draft models.** Speculative decoding relies on a draft that closely approximates the target model, yet such drafts are often unavailable. Early approaches assume smaller variants from the same model family (Leviathan et al., 2022), an assumption that frequently fails. For example, Kimi K2 (Team et al., 2025) was released without smaller counterparts, and even popular models like Qwen3 often lack matching drafts.

**Cause 2: Insufficient quality of open-source drafts.** While prior work has released a small number of draft checkpoints compatible with production inference engines (Li et al., 2025; 2024b), these are typically trained on limited, research-oriented datasets, constraining robustness and production readiness.

**Cause 3: Lack of robust training infrastructure.** Training high-quality drafts is non-trivial, requiring architectural customization and procedures such as TTT (Li et al., 2025). Existing implementations are largely ad hoc and do not scale across target models ranging from billions to trillion parameters (eag, 2025), hindering widespread adoption.

To address these challenges, we present `SpecForge`, a unified, production-oriented framework for training draft models for speculative decoding, with native support for advanced methods such as EAGLE-3. Unlike prior approaches that couple target and draft models under a uniform parallelization strategy, `SpecForge` adopts hybrid parallelism via target–draft decoupling, separating the frozen, inference-dominated target model from the lightweight, trainable draft model. This design applies inference-oriented parallelism to the target model and training-optimized parallelism to the draft model, and uniquely integrates a production-grade inference engine (e.g., SGLang) directly into the training pipeline. Together, these design choices improve scalability, reduce communication overhead, and enable efficient training against target models ranging from billions to over a trillion parameters. In addition, `SpecForge` exploits sparsity in tree-based attention and redundancy in TTT to optimize EAGLE-3 training, significantly reducing memory and compute consumption and achieving up to 9.9× end-to-end training speedup.

To enrich the open ecosystem, we release `SpecBundle`, a comprehensive suite of production-grade draft models for major open-source LLM families, including Llama-3/4, Qwen-3, GPT-OSS, Kimi K2, and Ling-Flash, trained on large, diverse corpora tailored for speculative decoding. `SpecBundle` delivers up to 4.8× speedup over non-speculative inference and up to 1.3× speedup over existing open draft checkpoints across multiple domains.

Finally, through training `SpecBundle`, we systematically

explore the speculative decoding design space and distill practical training recipes spanning draft architectures, dataset quality, and TTT configuration, providing actionable guidance for building high-quality drafts and deploying speculative decoding in production.

We summarize our main contributions as follows:
- We introduce `SpecForge`, a scalable and efficient framework for training speculative decoding draft models that combines hybrid parallelism via target–draft decoupling with optimized TTT training, achieving up to 9.9× end-to-end training speedup.
- We release `SpecBundle`, a suite of high-quality draft models for mainstream open-source LLMs, delivering stronger accuracy and up to 1.3× speedup over existing open checkpoints.
- We provide a systematic analysis of speculative decoding training recipes and design choices, offering practical guidance for real-world deployment and future research.

## 2. Preliminaries

### 2.1. Speculative Decoding

Speculative decoding (Chen et al., 2023; Leviathan et al., 2022; Xia et al., 2023; Miao et al., 2024; Cai et al., 2024; Li et al., 2024b;a; 2025; Hu et al., 2025) is a leading algorithmic approach for alleviating the memory-bound inefficiency of autoregressive LLM inference without retraining the target model or degrading generation quality. First formalized by Leviathan et al. (2022), it restructures inference by replacing serial token generation with parallel verification of candidate sequences. By employing a computationally inexpensive "draft model" to propose short sequences of tokens (speculation), speculative decoding allows the massive "target model" to verify these proposals in a single forward pass. This effectively converts the sequential generation problem into a batch processing problem, thereby increasing arithmetic intensity and better utilizing the massive parallel compute capabilities of modern hardware.

### 2.2. Theoretical Speedup

The efficiency of speculative decoding reflects a trade-off between tokens accepted from the draft model and the cost of generating them. Let $\alpha$ denote the per-token acceptance rate, $\gamma$ the number of speculative draft tokens, and $c$ the cost ratio between the draft model and the target model ($c = C_q/C_p$). The theoretical walltime speedup $S$ is defined as the ratio of standard autoregressive time to speculative decoding time. The speedup is given by:

$$S = \frac{E[\text{tokens}]}{1 + \gamma c} = \frac{1 - \alpha^{\gamma+1}}{(1-\alpha)(1+\gamma c)}$$

This formulation highlights three key factors: (i) high draft quality is critical, as $\alpha$ is fundamentally constrained by the KL divergence between draft and target distributions; (ii) the draft model must be substantially cheaper than the target ($c \ll 1$), otherwise draft overhead dominates; and (iii) $\gamma$ must be carefully chosen to balance increased parallelism against linear growth in draft cost.

## 2.3. EAGLE

EAGLE is a family of training-based speculative decoding methods that progressively improve draft model quality by aligning training with inference-time verification. EAGLE-1 (Li et al., 2024b) introduces the core idea of training a lightweight draft model to predict multiple future tokens whose correctness is efficiently verified by a large target model, enabling lossless acceleration over standard autoregressive decoding. EAGLE-2 (Li et al., 2024a) strengthens this framework by improving training stability and acceptance rates through refined objectives and better alignment between draft predictions and target verification. EAGLE-3 (Li et al., 2025) further advances the approach with *Training-Time Test (TTT)*, explicitly simulating multi-step autoregressive behavior during training, which significantly reduces error accumulation and yields substantially higher acceptance lengths and throughput in practice, making speculative decoding more effective and robust for large-scale deployment.

## 2.4. Training-Time Test

EAGLE-3 (Li et al., 2025) is the state-of-the-art speculative decoding method thanks to TTT, which simulates multi-step autoregressive generation during training to reduce error accumulation. At each TTT step, the model attends to a growing context consisting of the original training prefix and representations generated in previous steps, as shown in Algorithm 3 in the appendix.

For training prefix length $T$ and position $t > T$, tokens $1{:}T$ correspond to ground-truth inputs, while $T{+}1{:}t{-}1$ are generated during prior TTT steps. The attention output is computed over the concatenation of prefix and predicted keys and values. Its computation at step $t$ is as follows:

$$o_t = \text{softmax}\left(\frac{q_t \begin{bmatrix} K^{\text{train}} \\ K^{\text{pred}} \end{bmatrix}^{\top}}{\sqrt{d_k}}\right) \begin{bmatrix} V^{\text{train}} \\ V^{\text{pred}} \end{bmatrix},$$

where $d_k$ denotes the key dimensionality.

## 3. Challenges

Despite rapid progress in speculative decoding—particularly with EAGLE-3—the problem of training draft models has received comparatively little attention. Unlike conventional large-scale model training with frameworks such as Megatron (Narayanan et al., 2021) and DeepSpeed (Rasley et al., 2020), EAGLE-3 training exhibits a distinct systems profile: the draft model is lightweight, often consisting of a single Transformer layer, while the target model may contain billions to trillions of parameters. This asymmetry introduces unique challenges that make efficient and scalable draft-model training non-trivial.

**Rigid Parallelism Strategies.** Existing open-source implementations (eag, 2025; mod, 2025) couple the target and draft models into a unified module and apply a uniform parallelization strategy—typically fully sharded data parallelism (FSDP) (Zhao et al., 2023; Rasley et al., 2020). While this simplifies implementation, it is suboptimal from a systems perspective due to the substantial scale mismatch between the two models: the draft model is small and training-oriented, whereas the target model is large and inference-dominated. Although ZeRO-style sharding (Rasley et al., 2020) is effective for large-model training, it is poorly suited for inference-heavy workloads such as hidden-state generation. In contrast, high-performance inference engines favor tensor parallelism, expert parallelism, kernel fusion, and other inference-optimized techniques. Enforcing a single parallelization strategy therefore restricts inference-oriented optimizations and limits scalability and performance.

**Sub-optimal Prefill Performance.** During EAGLE-3 training, the target model processes the full input sequence to generate hidden states, which is equivalent to the prefill phase in standard LLM inference and dominates training cost for large target models. However, most existing EAGLE-3 training pipelines rely on naïve model implementations, either custom or directly imported from Hugging Face, that are optimized for general-purpose training rather than high-throughput inference. Consequently, they fail to exploit inference-specific optimizations such as efficient attention kernels, model-aware parallelism, optimized memory management, and CUDA Graph execution. This makes prefill a major bottleneck in draft-model training, inflating both training time and resource consumption, and motivates rethinking the training architecture to better align with inference-optimized execution.

## 4. SpecForge Methodologies

To address the above challenges, `SpecForge` introduces a set of system-level techniques that optimize draft-model training by decoupling execution paths and aligning each component with its most suitable runtime.

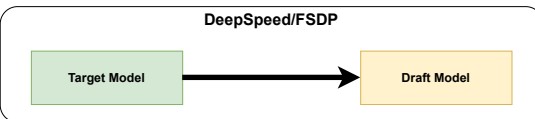

*(a)* Existing implmenetation wraps both the target model and draft model into a single parallel strategy

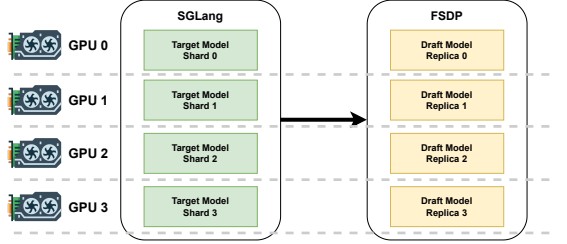

*(b)* `SpecForge` decouples the target model and draft model with hybrid parallelism

*Figure 1.* Architecture comparisons

---

**Algorithm 1** BlockMask Construction for TTT

**Input:** batch index $b$, query index $q_i$, key/value index $kv_i$, prefix length $Q_{\text{LEN}}$, sequence length $T$
**Output:** BlockMask $M$

*// Causal mask*
$m_{\text{causal}} \leftarrow (q_i \geq kv_i)$
$m_{\text{pad}} \leftarrow (kv_i < T)$
$M_{\text{causal}} \leftarrow m_{\text{causal}} \wedge m_{\text{pad}}$

*// Suffix mask*
$m_{\text{suffix}} \leftarrow (kv_i \geq Q_{\text{LEN}})$
$m_{\text{pad}} \leftarrow (kv_i \bmod Q_{\text{LEN}} < T)$
$m_{\text{diag}} \leftarrow ((kv_i - q_i) \bmod Q_{\text{LEN}} = 0)$
$M_{\text{suffix}} \leftarrow m_{\text{suffix}} \wedge m_{\text{pad}} \wedge m_{\text{diag}}$

*// BlockMask*
$M \leftarrow M_{\text{causal}} \vee M_{\text{suffix}}$

---

## 4.1. Target-Draft Decoupling

The original EAGLE-3 design tightly couples the target and draft models into a single parallelized module (Figure 1a). While convenient, this coupling fundamentally conflates two workloads with very different objectives and system requirements. Training frameworks are optimized for gradient computation and parameter updates, whereas inference engines prioritize throughput, latency, and memory efficiency. A unified execution model prevents both components from operating in their optimal regimes.

`SpecForge` **therefore adopts *target–draft decoupling* as a first-class abstraction.** The small draft model is trained using mature distributed training frameworks such as Deep-Speed (Rasley et al., 2020) and FSDP (Zhao et al., 2023), while the target model is executed exclusively through a high-performance inference SGLang (Zheng et al., 2023), as shown in Figure 1b. This is the **first attempt** to integrate production-grade inference engine for speculative decoding training. This separation enables independent optimization of execution backends and parallelization strategies, while also allowing trained draft models to be deployed directly in production inference pipelines without modification.

With such decoupled design, we can easily support *disaggregated training*, where the target and draft models are placed on independent GPUs. This enables flexible allocation of GPU resources for optimal performance. We leave this to future development.

### 4.1.1. HYBRID PARALLELISM

`SpecForge` applies distinct parallelization strategies to the draft and target models and explicitly decouples their data handling. The draft model is lightweight (typically 3–5%

of the target size), so we avoid tensor or pipeline parallelism and instead shard only optimizer states and gradients, equivalent to ZeRO-2 in DeepSpeed (Rasley et al., 2020), to minimize communication overhead during training.

The target model is executed using SGLang's inference runner and may employ a mixture of data parallelism (DP), tensor parallelism (TP), and expert parallelism (EP). To accommodate this heterogeneity, `SpecForge` organizes data from the perspective of a *target model instance*. Each target model instance prepares hidden states locally for the draft models co-located on the same devices. Specifically, a target model instance produces $M \times N$ data samples per step, where $M$ denotes the TP/EP parallel degree and $N$ is the number of draft models residing on the same device. Each draft model then fetches and consumes a disjoint $1/N$ shard of the generated hidden states for training, avoiding cross-device communication.

This decoupled data mapping enables flexible deployment of target and draft models—either co-located or disaggregated—while preserving locality and scalability across mixed parallelism configurations. Unless otherwise noted, all experiments in this paper use the co-located setting.

## 4.2. TTT Optimization

Beyond parallelization, we analyze the memory behavior of Training-Time Test (TTT) and observe that a TTT length of 7 incurs substantial GPU memory overhead. We identify attention activations as the dominant contributor and introduce two complementary optimizations to significantly reduce training memory consumption.

---

**Algorithm 2** In-place Backward for Log-Softmax Loss

---

**Input:** logits $z$, target $p$, upstream gradient $g$
**Output:** gradient w.r.t. logits (stored in $z$)
$s \leftarrow \sum_i (p_i g_i)$
$\pi \leftarrow \mathrm{softmax}(z)$
$z \leftarrow -(p \odot g - \pi \odot s)$

---

### 4.2.1. SPARSE TREE ATTENTION

In the naive TTT implementation, attention logits are materialized and stored as intermediate activations. As TTT unrolls multiple autoregressive steps, these logits accumulate and dominate memory usage. Our profiling shows that they account for up to **80%** of the total activation memory.

To mitigate this bottleneck, we implement sparse tree attention using *FlexAttention*. As shown in Figure 7, generated tokens do not attend to all previous tokens. Instead, they only attend to tokens in the previous nodes in the candidate tree. We construct a BlockMask that encodes the tree-structured attention pattern (Algorithm 1). At each TTT step, we update the KV cache and invoke FlexAttention with the corresponding mask, substantially reducing activation memory.

### 4.2.2. MEMORY-EFFICIENT GRADIENT COMPUTATION

We further observe that loss computation introduces nontrivial memory overhead, as EAGLE-3 evaluates the loss at every training-time test (TTT) step, producing multiple activation and gradient tensors. To reduce this overhead, we implement a custom Triton kernel for the masked softmax backward pass (Algorithm 2) that performs in-place gradient computation. Specifically, since logits are no longer needed after the forward pass, our kernel overwrites the logits tensor with its gradients during backpropagation, eliminating separate gradient buffers and significantly reducing memory usage during TTT training.

## 5. Evaluation

### 5.1. Experimental Setup

We evaluate the performance of `SpecForge` against existing EAGLE3 training implementations. We consider two publicly available codebases: (1) the official implementation released by SafeAILab alongside the EAGLE3 paper (eag, 2025), and (2) a third-party implementation developed by NVIDIA's Model Optimizer team (mod, 2025). As both adopt a similar monolithic design that jointly wraps target and draft models within DeepSpeed, we select the official SafeAILab implementation as the representative baseline.

All experiments are conducted on a cluster of eight NVIDIA H200 GPUs with a sequence length of 4096. For each method, the batch size is tuned to maximize training throughput under GPU memory constraints.

### 5.2. End-to-end Performance

We conduct end-to-end training experiments on four models spanning different scales and architectures: LLaMA3.1-8B, LLaMA3.3-70B, Qwen3-30B-A3B, and Qwen3-235B-A22B. Training throughput is measured in tokens per second. For `SpecForge`, we enable tensor parallelism (TP), FlashAttention kernels, and CUDA Graphs, with the tensor-parallel degree chosen based on the target model scale. The draft model is trained using ZeRO Stage 2 for memory-efficient data-parallel execution. For the baseline, we evaluate both ZeRO Stage 2 and ZeRO Stage 3 configurations and report the best-performing result.

Table 1 summarizes the results. `SpecForge` consistently outperforms the baseline across all model scales, achieving up to a $9.99\times$ speedup. The baseline performance degrades sharply as the model size increases, primarily due to the following factors:

- Under ZeRO Stage 2, although optimizer states and gradients are sharded, the frozen target model parameters remain fully replicated on each device, resulting in poor scalability for large models.
- ZeRO Stage 3 additionally shards model parameters, but frequent all-gather operations during target-model inference incur substantial communication overhead, which severely limits throughput.

These results highlight the benefits of target–draft decoupling. For large-scale models such as Qwen3-235B-A22B, ZeRO-stage3 leads to communication-dominated execution and extremely low throughput. In contrast, `SpecForge` avoids unnecessary weight gathering, enabling 9.9x speedup over the baseline. The results also underscore the importance of integration with a mature inference engine. As demonstrated by the LLaMA3.1-8B experiments, even when neither the baseline nor `SpecForge` parallelizes the target model, `SpecForge` still achieves a $2.01\times$ speedup due to the highly optimized prefill execution provided by SGLang.

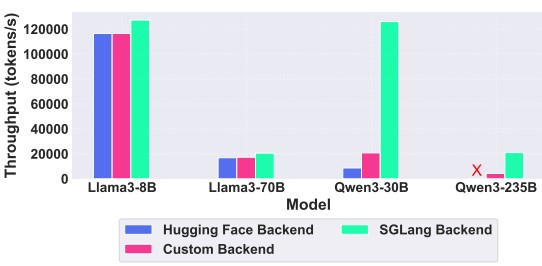

*Figure 2.* Training time with different execution backends

| Target Model | Framework | Target Model | Draft Model | Max Batch Size | Step Time (s) | Throughput (tokens/s) | speedup |
|---|---|---|---|---|---|---|---|
| Llama3.1-8B | Baseline | ZeRO 2 | ZeRO 2 | 16 | 1.04 | 63015.4 | 1 |
| | SpecForge | TP=1 | ZeRO 2 | 64 | 2.07 | **126639.6** | **2.01** |
| Llama3.3-70B | Baseline | ZeRO 2 | ZeRO 2 | 16 | OOM | - | - |
| | Baseline | ZeRO 3 | ZeRO 3 | 8 | 2.21 | 14827.1 | 1 |
| | SpecForge | TP=4 | ZeRO 2 | 16 | 3.18 | **20608.8** | **1.39** |
| Qwen3-30B-A3B | Baseline | ZeRO 2 | ZeRO 2 | 8 | 1.12 | 29257.1 | 1 |
| | Baseline | ZeRO 3 | ZeRO 3 | 8 | 5.07 | 6463.1 | 0.2 |
| | SpecForge | TP=4 | ZeRO 2 | 16 | 0.52 | **126030.8** | **4.31** |
| Qwen3-235B-A22B | Baseline | ZeRO 2 | ZeRO 2 | 8 | OOM | - | - |
| | Baseline | ZeRO 3 | ZeRO 3 | 8 | 11.2 | 2025.7 | 1 |
| | SpecForge | TP=8 | ZeRO 2 | 8 | 1.62 | **20227.2** | **9.99** |

*Table 1.* End-to-end performance on various models.

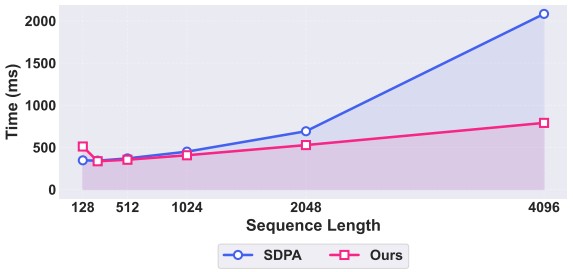

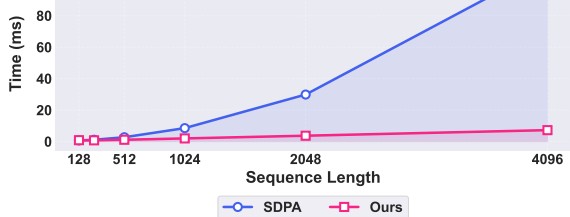

*(a)* Kernel wall time for attention

*(b)* Peak memory consumption for attention

*Figure 3.* Comparison of execution time and memory usage between naive EAGLE3 attention and our optimized kernel.

## 5.3. Ablation Studies

### 5.3.1. IMPACT OF TARGET MODEL BACKENDS

We further delve into the importance of integration with productin-grade inference engine. `SpecForge` supports three backends: a Hugging Face backend that reuses Transformers implementations with built-in tensor parallelism when available; an SGLang backend that leverages SGLang's built-in optimization; and a custom backend for manual model implementation.

We evaluate all execution backends using the same models and tensor-parallel settings as in Table 1. As shown in Figure 2, SGLang consistently outperforms the other backends, delivering up to 6.8× higher training throughput. This gap highlights the difficulty of optimizing the prefill stage, especially for MoE models: in the Qwen3 experiments, both the custom and Hugging Face backends achieve substantially lower throughput, with the latter exhibiting runtime failure at scale. These results re-affirms the importance of a mature, inference-optimized backend for scalable EAGLE3 training.

### 5.3.2. IMPACT OF TTT OPTIMIZATION

We micro-benchmark our optimized attention kernel against a native SDPA implementation, measuring execution time and peak memory usage at the final TTT step with TTT length set to 7. As shown in Figure 3, the optimized kernel

significantly reduces both metrics, achieving 62.1% lower execution time and 93.5% lower peak memory usage at a sequence length of 4096 on a single NVIDIA H200 GPU. The performance gap further widens with longer sequences, demonstrating the kernel's effectiveness for long-context EAGLE3 training.

For the log softmax loss, we conduct a micro-benchmark with a sequence length of 4096 and a vocabulary size of 32,000. Our optimized kernel reduces peak memory usage from 7.32 GB to 5.62 GB—an improvement of 23.3%—while incurring no measurable performance degradation.

### 5.3.3. IMPACT OF KERNEL PRECISION

An exact bitwise match between a torch-native kernel and our Triton kernel is generally infeasible because floating-point operations are non-associative and Triton and torch operators may use different intermediate precisions. Rather than relying on bitwise equivalence, we validate numerical correctness with tests that check our kernel output against the torch kernel within an acceptable tolerance. We also verify training stability and inference consistency for Llama3-8B. As shown in the loss curve in Figure 4a, where the native kernel and our kernel deliver closely aligned curves. In addition, models trained with the baseline and our kernel achieve comparable performance across all benchmarks reported in

| Target Model | Framework | Target Model | Draft Model | Max Batch Size | Step Time (s) | Throughput (tokens/s) | speedup |
|---|---|---|---|---|---|---|---|
| Llama-3.1-8B | Baseline | ZeRO 2 | ZeRO 2 | 1 | 0.53 | 7728.30 | 1 |
| | SpecForge | TP=2 | ZeRO 2 | 2 | 0.87 | 9416.09 | **1.22** |
| Qwen3-30B-A3B | Baseline | ZeRO 3 | ZeRO 3 | 8 | 0.61 | 6714.75 | 1 |
| | SpecForge | TP=4 | ZeRO 2 | 16 | 0.31 | 13212.91 | **1.97** |

*Table 2.* Training Performance on H100 GPUs

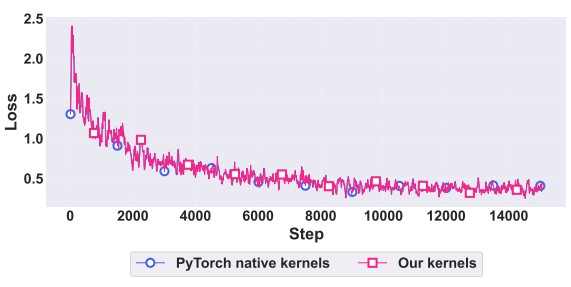

*(a)* Training Loss for TTT step = 7

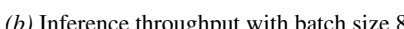

*(b)* Inference throughput with batch size 8

*Figure 4.* Impact of kernel precision on training and inference for Llama-3.1-8B using SpecForge

Figure 4b, further validating the numerical correctness of our kernels.

### 5.3.4. IMPACT OF GPU DIVERSITY

To assess robustness across GPU architectures, we conduct additional experiments on NVIDIA H100 GPUs using Llama-3.1-8B and Qwen3-30B-A3B, as shown in Table 2. For each model, we use the largest batch size that avoids out-of-memory errors. Our method preserves the same performance trends on H100 and achieves up to a 2× speedup, showing that the gains are not tied to a single GPU platform.

The speedup on H100 is lower than on H200 mainly because H100's smaller memory capacity limits the maximum per-GPU training batch size, reducing exploitable parallelism. This is a hardware constraint rather than an algorithmic limitation. In practice, memory pressure can be mitigated by placing the draft and target models on separate GPUs, enabling larger effective batch sizes and better scalability.

## 6. SpecBundle

As part of our open-source effort, we train EAGLE-3 draft models for several mainstream open-source LLMs, including Llama, Qwen, and Kimi, released collectively as SpecBundle. These drafts are trained on the Open-PerfectBlend dataset (Xu et al., 2024), which contains 1.4M balanced conversations spanning chat, math, coding, and instruction-following domains. To improve draft quality, we regenerate assistant responses using the target model with a temperature of 0.8 and train the drafts from scratch on the regenerated data for two epochs using a cosine-annealed learning rate schedule with an initial rate of $10^{-4}$.

### 6.1. Evaluation Results

We evaluate SpecBundle on a diverse set of benchmarks covering mathematics (Math500 (Lightman et al., 2023), GSM8K (Cobbe et al., 2021)), coding (HumanEval (Chen et al., 2021), LiveCodeBench (Jain et al., 2024)), and other domains (GPQA (Rein et al., 2024), MTBench (Chen et al., 2025a), FinanceQA (Mateega et al., 2025)). Results for mathematics and coding are reported in Table 3, with remaining results summarized in Table 5 in the appendix.

We evaluate SpecBundle using SGLang on NVIDIA H200 GPUs, comparing it against (1) standard non-speculative inference and (2) speculative decoding with existing open-source draft models, where available. Baseline drafts include EAGLE-3 checkpoints released by the original authors (Li et al., 2025) and the LMSYS team, although draft availability remains limited for many target models. We fix the concurrency to 1 for LLaMA-3.1-8B and 8 for larger models, applying tensor parallelism according to model scale. We set the temperature to 0 and evaluate multiple speculative decoding configurations by varying the number of drafting steps, draft tokens, and drafting top-$k$.

We report the highest achieved throughput. On the coding and mathematics benchmarks, SpecBundle achieves speedups over baselines ranging from 1.61× to 4.48× (Table 3). This performance gap arises because existing checkpoints are primarily trained on the ShareGPT and Ultra-Chat datasets, which contain limited coverage of math- and code-centric samples. These results underscore the critical role of data composition in training a well-balanced and high-performing draft model. Similar performance can be observed on other domain benchmarks as shown in Table 5

| Target Model | Draft Model | #GPUs | LiveCodeBench | | HumanEval | | GSM8K | | Math500 | |
|---|---|---|---|---|---|---|---|---|---|---|
| | | | Throughput | Speedup | Throughput | Speedup | Throughput | Speedup | Throughput | Speedup |
| Llama-3.1-8B | - | 1 | 189.7 | 1 | 190.9 | 1 | 181.8 | 1 | 191.0 | 1 |
| | Existing | | 398.4 | 2.10 | 480.3 | 2.52 | 228.6 | 1.26 | 422.4 | 2.21 |
| | SpecBundle | | **516.9** | **2.72** | **571.5** | **2.99** | **329.7** | **1.81** | **638.0** | **3.34** |
| Llama-3.3-70B | - | 4 | 560.9 | 1 | 561.0 | 1 | 453.2 | 1 | 567.4 | 1 |
| | Existing | | 1303.4 | 2.32 | 1282.8 | 2.29 | 521.5 | 1.15 | 1122.2 | 1.98 |
| | SpecBundle | | **1459.4** | **2.60** | **1506.0** | **2.68** | **722.0** | **1.59** | **1524.9** | **2.69** |
| Llama-4-Scout | - | 8 | 484.3 | 1 | 631.9 | 1 | 455.9 | 1 | 561.8 | 1 |
| | Existing | | 1601.3 | 3.31 | 1556.5 | 2.46 | 816.6 | 1.79 | 1479.0 | 2.63 |
| | SpecBundle | | **2170.2** | **4.48** | **1944.8** | **3.08** | **971.9** | **2.13** | **2110.3** | **3.76** |
| Qwen-30B-A3B | - | 4 | 1492.6 | 1 | 1366.6 | 1 | 1071.3 | 1 | 1469.0 | 1 |
| | SpecBundle | | **3413.0** | **2.29** | **3070.0** | **2.25** | **1499.6** | **1.40** | **3636.1** | **1.48** |
| Qwen-235B-A22B | - | 8 | 598.2 | 1 | 553.1 | 1 | 469.1 | 1 | 587.4 | 1 |
| | Existing | | 803.8 | 1.34 | 889.9 | 1.61 | 697.0 | 1.49 | 821.8 | 1.39 |
| | SpecBundle | | **1155.7** | **1.93** | **1267.5** | **2.29** | **758.3** | **1.62** | **1399.2** | **2.38** |
| Ling-Flash-V2 | - | 8 | 770.4 | 1 | 740.2 | 1 | 674.3 | 1 | 762.7 | 1 |
| | SpecBundle | | **1366.4** | **1.77** | **1359.0** | **1.83** | **1323.0** | **1.96** | **1685.6** | **2.21** |
| Kimi-K2 | - | 8 | 500.1 | 1 | 466.1 | 1 | 337.9 | 1 | 492.1 | 1 |
| | SpecBundle | | **904.4** | **1.81** | **897.9** | **1.93** | **544.2** | **1.61** | **1022.7** | **2.08** |

*Table 3.* Performance of various models on math and coding benchmarks

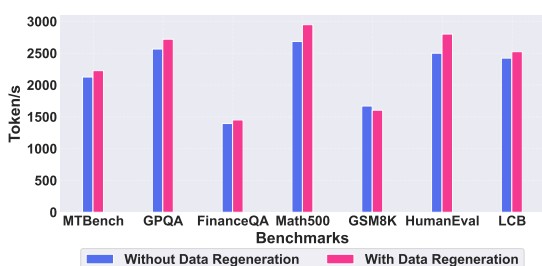

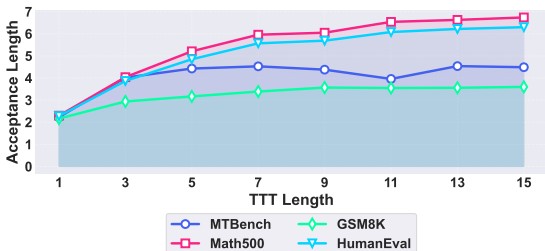

*Figure 5.* Inference performance of Llama3.1-8B with EAGLE3 trained on datasets with and without regenerating the responses. The experiment was conducted on 1 H200 GPU with batch size 8.

*Figure 6.* Scaling TTT for the Llama3.1-8B model on the perfect-blend dataset.

in Appendix.

SpecBundle enriches the open-source ecosystem with a broader supply of draft models and delivers substantial performance improvements for production-grade inference. While the current release focuses on instruct models, we plan to extend support to reasoning models and vision–language models in future iterations.

## 7. Training Insights

Through SpecBundle, we derive some key insights which provides practical guidance for training.

### 7.1. Impact of Data Regeneration

Prior work suggests that EAGLE methods are largely insensitive to training data and therefore recommends training on the original dataset to reduce computational cost (Li et al., 2024b). Our results indicate that this assumption does not

always hold. We train EAGLE-3 draft models for LLaMA-3.1-8B using both the original PerfectBlend dataset and a regenerated variant. As shown in Figure 5, data regeneration consistently improves performance across nearly all benchmarks, with FinanceQA as the sole exception, and yields an average throughput gain of 5.3%. The acceptance length results can be found in Figure 8 in appendix. Although the absolute improvement is modest, such gains can translate into substantial cost savings at scale, given the widespread deployment of speculative decoding in online inference systems.

### 7.2. Impact of Training-Time Test

In the original EAGLE-3 implementation (eag, 2025; Li et al., 2025), the training-time test (TTT) length is fixed to 7. To study its impact, we vary the TTT length from 1 to 15 and evaluate inference performance. As shown in Figure 6, increasing the TTT length consistently improves the acceptance length across most benchmarks. However, the optimal TTT length is task-dependent: on MT-Bench, a length of 7 already achieves strong performance, whereas more

| Draft Model | Same Params as Dense? | #Experts | topk | Expert Intermediate size | MTBench | Math500 | GSM8K | HumanEval |
|---|---|---|---|---|---|---|---|---|
| Dense | - | - | - | - | **2.53** | **2.78** | 2.25 | **2.91** |
| MoE | Yes | 64 | 8 | 224 | 2.49 | 2.35 | 2.20 | 2.32 |
| | Yes | 32 | 8 | 448 | 2.49 | 2.35 | **2.30** | 2.55 |
| | No, 8 times more | 64 | 8 | 1792 | 2.46 | 2.40 | 2.20 | 2.44 |

*Table 4.* Acceptance length of MoE models with different settings. The results are obtained with configurations MoE top-k = 1, EAGLE-3 number of steps = 3, EAGLE3 top-k = 1 and EAGLE3 number of draft tokens = 4.

challenging and longer-horizon tasks—such as Math500, GSM8K, and HumanEval—benefit from larger TTT lengths of approximately 15.

Higher acceptance does not translate proportionally into higher output throughput, as longer TTT increases the cost of the drafting phase; for example, TTT = 11 achieves higher throughput (3425 tokens/s) than TTT = 15 (3243 tokens/s). Moreover, increasing TTT length proportionally raises both training time and memory consumption, leading to a clear performance–efficiency trade-off. In practice, we recommend selecting TTT via small-scale scaling studies prior to full training; Adaptive, sample-dependent TTT strategies may further reduce training cost and are left for future work.

### 7.3. Choice of Draft Models

Recent large language models increasingly adopt Mixture-of-Experts (MoE) architectures for their favorable performance–efficiency trade-off (DeepSeek-AI et al., 2024; Team et al., 2025). However, existing EAGLE-3 draft models remain dense, and the applicability of MoE to speculative decoding is largely unexplored. To address this gap, we train dense and MoE draft models on ShareGPT and evaluate them across a range of benchmarks.

The results are summarized in Table 4. Overall, dense draft models consistently outperform their MoE counterparts, with particularly large margins on Math500 and HumanEval. These results indicate that dense architectures are generally more effective for draft modeling in speculative decoding. We attribute the inferior performance of MoE drafts to the limited training data available in this setting: unlike large-scale pretraining, speculative decoding datasets provide insufficient tokens per expert, leading to under-trained and weak individual experts. Furthermore, MoE training becomes especially unstable when the number of experts is small. For instance, the 4-expert configuration performs poorly, achieving an acceptance length of only 1.81 on MT-Bench.

### 8. Related Work

There are several existing implementations focusing on speculative decoding draft model training projects including EAGLE-3 official release (eag, 2025), Nvidia Model Op-

timizer (mod, 2025) and speculators (Hat, 2025) and they all adopted the same approach by wrapping both draft and target model into a single module for traninig, which delivers undesired performance as verified by our experiments. In comparison, `SpecForge` offers a decoupled architecture for flexibility and individual optmization for draft and target model and optimized the TTT training specifically, outperforming the baseline implementations.

A separate but relatively limited line of work studies algorithmic techniques for improving the training efficiency of EAGLE-3. For example, FastEAGLE (Li et al., 2026) redesigns the training curriculum and restructures the offline training pipeline to reduce both training time and storage overhead. These approaches are orthogonal to `SpecForge`: while they modify the training algorithm or data pipeline, `SpecForge` focuses on system-level optimization and preserves the original EAGLE-3 training workflow. As a result, `SpecForge` is fully compatible with such algorithmic advances.

Recent work has also explored new speculative decoding paradigms beyond standard EAGLE-style draft models. For instance, DFlash (Chen et al., 2026) improves speculative decoding latency by adopting diffusion language models. The architecture of `SpecForge` is designed to be extensible, and our latest codebase supports integration with such methods. In addition, methods such as ReSpec (Chen et al., 2025b) use EAGLE-3 to accelerate the rollout phase of reinforcement learning training. Unlike `SpecForge`, these approaches do not target draft-model training efficiency; instead, they focus on improving end-to-end reinforcement learning throughput.

### 9. Conclusion

In summary, We present `SpecForge`, a high-performance framework for training speculative decoding draft models. By combining target–draft decoupling with optimized TTT, `SpecForge` significantly reduces memory usage and boosts training throughput, achieving up to 9.9× speedup over existing methods. We also release SpecBundle, a suite of production-ready EAGLE-3 draft models, along with systematic analyses to support real-world training tasks.

## Impact Statement

his paper presents work whose goal is to advance the field of machine learning. There are many potential societal consequences of our work, none of which we feel must be specifically highlighted here.

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

# A. Training-time Test

## A.1. TTT Training Algorithm

---

**Algorithm 3** TTT Attention

---

**Input:** query $q_t$, prefix keys $K^{\text{train}}$, prefix values $V^{\text{train}}$, cached keys $\{k_i\}_{i>T}$, cached values $\{v_i\}_{i>T}$
**Output:** attention output $o_t$
$S \leftarrow \frac{q_t (K^{\text{train}})^\top}{\sqrt{d_k}}$

**for** $i = T + 1$ **to** $t - 1$ **do**
    $s_i \leftarrow \frac{q_t \cdot k_i}{\sqrt{d_k}}$
    $S \leftarrow \text{concat}(S, s_i)$
**end for**

$\alpha \leftarrow \text{softmax}(S)$
$o_t \leftarrow \alpha \cdot V^{\text{train}}$

**for** $i = T + 1$ **to** $t - 1$ **do**
    $o_t \leftarrow o_t + \alpha_i v_i$
**end for**

---

## A.2. TTT Mask

The attention mask used in TTT training is illustrated in Figure 7. The resulting attention pattern follows a tree-structured layout: the prefix tokens form a lower-triangular mask, while the tokens generated at each TTT step appear along diagonal bands.

| key / query | Prefix Sequence | | | | Step 1 | | | | Step 2 | | | |
|---|---|---|---|---|---|---|---|---|---|---|---|---|
| | How | can | ... | \<pad\> | are | we | ... | \<pad\> | you | help | ... | \<pad\> |
| you | 1 | 0 | 0 | 0 | 1 | 0 | 0 | 0 | 1 | 0 | 0 | 0 |
| help | 1 | 1 | 0 | 0 | 0 | 1 | 0 | 0 | 0 | 1 | 0 | 0 |
| ... | 1 | 1 | 1 | 0 | 0 | 0 | 1 | 0 | 0 | 0 | 1 | 0 |
| \<pad\> | 0 | 0 | 0 | 0 | 0 | 0 | 0 | 0 | 0 | 0 | 0 | 0 |

*Figure 7.* EAGLE3 attention mask used in Training-Time Testing.

## B. SpecBundle Performance

The performance of SpecBundle models on non-math and non-coding benchmarks is summarized in Table 5. The results show that SpecBundle consistently outperforms existing draft models by a substantial margin. For example, SpecBundle achieves up to a 34% throughput improvement compared to prior draft model checkpoints.

| Target Model | Draft Model | #GPUs | MTBench | | GPQA | | FinanceQA | |
|---|---|---|---|---|---|---|---|---|
| | | | Throughput | Speedup | Throughput | Speedup | Throughput | Speedup |
| Llama-3.1-8B | - | 1 | 190.0 | 1 | 190.5 | 1 | 185.7 | 1 |
| | Existing | | **454.7** | 2.39 | 438.1 | 2.30 | 237.2 | 1.27 |
| | SpecBundle | | 450.0 | 2.37 | **514.2** | **2.70** | **258.6** | **1.39** |
| Llama-3.3-70B | - | 4 | 540.5 | 1 | 575.7 | 1 | 512.6 | 1 |
| | Existing | | **1272.7** | **2.35** | 1049.0 | 1.82 | 981.7 | 1.92 |
| | SpecBundle | | 1253.0 | 2.31 | **1405.1** | **2.44** | **1022.7** | **2.00** |
| Llama-4-Scout | - | 8 | 502.1 | 1 | 541.0 | 1 | 288.9 | 1 |
| | Existing | | 1253.0 | 2.50 | 1405.1 | 2.60 | 1022.7 | 3.54 |
| | SpecBundle | | **1312.4** | **2.61** | **1502.2** | **2.78** | **1189.6** | **4.12** |
| Qwen-30B-A3B | - | 4 | 1341.3 | 1 | 1410.4 | 1 | 1320.1 | 1 |
| | SpecBundle | | **2086.1** | **1.55** | **2341.3** | **1.66** | **1779.0** | **1.35** |
| Qwen-235B-A22B | - | 8 | 529.9 | 1 | 563.2 | 1 | 539.5 | 1 |
| | Existing | | 642.7 | 1.21 | 716.7 | 1.27 | 689.4 | 1.28 |
| | SpecBundle | | **814.5** | **1.54** | **826.5** | **1.47** | **889.0** | **1.65** |
| Ling-Flash-V2 | - | 8 | 728.5 | 1 | 794.1 | 1 | 747.7 | 1 |
| | SpecBundle | | **1022.6** | **1.40** | **1185.7** | **1.49** | **863.9** | **1.16** |
| Kimi-K2 | - | 8 | 430.9 | 1 | 505.4 | 1 | 433.4 | 1 |
| | SpecBundle | | **533.8** | **1.24** | **811.4** | **1.61** | **660.0** | **1.52** |

*Table 5.* Performance of various models on general benchmarks

## C. Results for Ablation Studies

### C.1. Impact of Data Regenration

Figure 8 reports the acceptance length across multiple benchmarks for models trained with and without data regeneration. All evaluations are performed under a fixed configuration with EAGLE-3 steps set to 3, top-$k$ equal to 1, and 4 draft tokens.

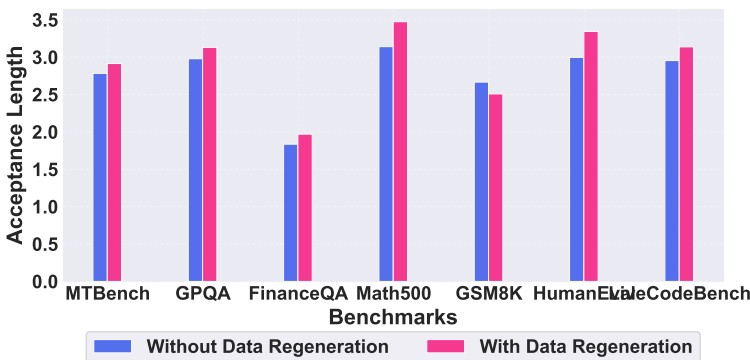

*Figure 8.* Inference performance of Llama3.1-8B with EAGLE3 trained on datasets with and without regenerating the responses. The experiment was conducted on 1 H200 GPU with batch size 8.

