# OpenReview forum: "SpecForge: A Flexible and Efficient Open-Source Training Framework for Speculative Decoding"
_ICML.cc/2026/Conference — ICML 2026 regular_

### Official Review · Reviewer_HZmt · 2026-03-07

**Soundness:** 3
**Presentation:** 4
**Significance:** 4
**Originality:** 3
**Overall Recommendation:** 4
**Confidence:** 3

**Summary:**

The paper presents SpecForge, a system-level framework designed to scale speculative decoding to frontier-class Large Language Models (LLMs) exceeding 200B parameters. The core contribution is a decoupled training architecture that separates the target model (inference engine) from the draft model (training engine), overcoming VRAM bottlenecks. To optimize this process, the authors introduce Algorithm 1: BlockMask Construction, which utilizes modular arithmetic to exploit the sparse tree structure of speculative drafts, reducing attention memory by 93.5%. The authors demonstrate significant throughput speedups (up to 4.48x) on models like Llama-4-Scout during inference.

**Compliance With Llm Reviewing Policy:**

Affirmed.

**Key Questions For Authors:**

- Bit-Identity Verification: Can the authors provide a bit-identity test (e.g., at $T=0$) comparing SpecForge outputs to a Eagle 3 baseline to ensure the custom Triton kernels and decoupling do not introduce divergence?
- Acceptance Length Comparison: To isolate the impact of Algorithm 1, please provide a table comparing the Acceptance Length of SpecForge versus vanilla EAGLE-3 on a common baseline (e.g., Llama-3-70B) across different draft lengths (k=4,8, and 16).
- Numerical Stability: How does the system handle potential precision loss when passing hidden states between the decoupled inference and training engines? Is there any requantization involved?
Robustness: Does the modulo-based mask construction support variable tree topologies, or is it strictly optimized for fixed-width/fixed-depth structures?

**Limitations:**

Yes. The authors discuss the hardware requirements and the specific dependency on Triton-supported backends.

**Strengths And Weaknesses:**

Strengths:
Significance: Addresses a critical bottleneck in LLM deployment—the memory cost of training draft models for very large targets. Scaling speculative decoding to 235B models is a high-impact practical achievement.
Soundness (System): The decoupling of engines is a robust solution to the "Memory Wall," allowing for parallelized, asynchronous training-time tests (TTT).

Originality:
While the distillation logic builds on the EAGLE family, the BlockMask modular arithmetic is a novel and clever hardware-aware optimization that enables the use of high-performance Triton kernels on sparse tree structures.

Weaknesses:
Originality (Model): There is a heavy conceptual overlap with EAGLE-3. The paper focuses primarily on the system for training rather than a new model architecture for speculation.

Soundness (Evaluation): The evaluation focuses almost exclusively on throughput. There is a lack of "Bit-Identity" testing to prove that the custom Triton kernels and decoupled synchronization do not introduce numerical divergence or logic errors.

The paper lacks a direct comparison of Acceptance Length ($L$) between SpecForge and the EAGLE-3 baseline, making it difficult to determine if the decoupled training improves or degrades the quality of the draft model. The paper only shows the impact of data generation in their experiment (fig 7).

---

> ### Author Rebuttal · Authors · 2026-03-31
>
> We thank the reviewers for their constructive feedback and insightful comments. We wish to address your concerns below.
>
> ## Originality
>
> We focus on improving the efficiency and scalability of the training system for speculative decoding, rather than introducing new decoding algorithms. Our discussion and evaluation center on EAGLE-3, as it represents the state of the art at the time of submission and provides a clear reference point for reviewers who may be less familiar with this line of work. Importantly, our system is designed as a general framework for target–draft decoupled training and is not tied to any single method. To demonstrate this flexibility, we integrate DFlash into SpecForge using its publicly available training pipeline, showing that our framework can support diverse approaches with minimal modification. We will include the full codebase in the supplementary materials to provide further validation after discussion period as we are not allowed to do so during this period.
>
> ## Kernel Precision
>
> It is not feasible to achieve an absolute bitwise match for a torch-native kernel and a kernel written in triton for mainly the following reasons:
>
> 1. Floating-point addition and multiplication are not associative and CUDA does not guarantee the reduction order
> 2. Different instruction selections: triton may select different low-level operations such as fma differently from the torch's native operator implementation
> 3. Triton and torch operators may use different intermediate precisions.
>
> Instead of achieving perfect bitwise match to demonstrate there is no loss in precision, we implemented unit tests to make sure that our kernel's output is within the acceptable tolerance range compared to the torch's kernel. Moreover, we verified that the kernel won't cause divergence by conducting training with the torch's native kernel and ours for the Llama3-8B model. In the loss curve figure in https://ibb.co/3mjnWFj8 , where the orange curve is for the baseline and the pink curve is for our system, we can observe that these two loss curves closely match each other. Meanwhile, the performance of the models trained with the baseline and ours show comparable performance on all benchmarks mentioned in the paper. **The result is given in the format "output-throughput (acceptance length)**
>
> | Model     | Setting                      | MTBench        | Math500        | GSM8K          | HumanEval      | LiveCodeBench  | GPQA           | FinanceQA      |
> |-----------|------------------------------|----------------|----------------|----------------|----------------|----------------|----------------|----------------|
> | Llama3-8B | Without Speculative Decoding | 1283.62 (1)    | 1343.99 (1)    | 1231.00 (1)    | 1348.83 (1)    | 1303.98 (1)    | 1318.24 (1)    | 1083.60 (1)    |
> | Llama3-8B | Baseline                     | 2335.49 (2.49) | 2905.37 (2.74) | 1998.38 (2.28) | 2867.89 (2.81) | 2686.98 (2.71) | 2972.41 (2.86) | 1635.20 (1.85) |
> | Llama3-8B | SpecForge                    | 2334.64 (2.49) | 2916.42 (2.75) | 2012.74 (2.32) | 2898.16 (2.84) | 2685.74 (2.71) | 3002.29 (2.86) | 1632.41 (1.83) |
>
> ## Acceptance Length Comparison
>
> We did not directly compare acceptance length because we evaluated multiple EAGLE-3 configurations and reported the best performance for both the baseline and our system. As configurations vary, a direct comparison of acceptance length is not always meaningful, since increasing the number of draft tokens typically leads to higher acceptance length. Nonetheless, we agree with the reviewer that such a comparison is valuable for demonstrating that our system preserves performance.
> In the table above, we report the acceptance metrics for both the baseline and our system under identical settings in SGLang (number of drafting steps = 3, top-k = 1, number of draft tokens = 4, concurrent requests = 8). The values in brackets denote the acceptance length. The results show that model quality is not degraded, and our system achieves closely matching acceptance behavior across a wide range of benchmarks.
>
>
> ## Precision Loss during Passing Hidden States
>
> There is no precision loss when passing hidden states between the inference and training engines as this process does not involve any computation. It is basically that each draft model will take the shard of the hidden states by splitting it in the batch dimension. Therefore, there is no need to requantization as well. We did not conduct any requantization as we use the same BF16 precision for the target model inference and draft model training. Even if the target and draft models are in different precisions, it won't incur a precision loss as long as we deploy the target and draft models with the same precision settings.

---

> > ### Author Rebuttal · Reviewer_HZmt · 2026-04-02
> >
> > Thanks for response and clarity. I maintain my order.

---

### Official Review · Reviewer_BSLo · 2026-03-12

**Soundness:** 4
**Presentation:** 3
**Significance:** 4
**Originality:** 4
**Overall Recommendation:** 5
**Confidence:** 4

**Summary:**

The paper introduces SpecForge, an efficient open-source training framework for speculative decoding, specifically optimized for EAGLE-3. To address the scarcity of high-quality draft models and the inefficiency of current training infrastructures, the authors propose a target-draft decoupling architecture and hybrid parallelism. By integrating production-grade inference engines (SGLang) and implementing specialized kernels for Training-Time Test (TTT), SpecForge achieves up to a 9.9x training speedup. The authors also release SpecBundle, a suite of pre-trained draft models that deliver up to 4.48x end-to-end inference acceleration.

**Compliance With Llm Reviewing Policy:**

Affirmed.

**Key Questions For Authors:**

1. Given that SpecForge's efficient kernels enable unrestricted scaling of data and context length, do the authors believe EAGLE-3's autoregressive modeling can fundamentally surpass native MTP in acceptance rates?

2. As "Block Diffusion" methods like DFlash begin to outperform EAGLE-3, are there plans to generalize this infrastructure to support non-autoregressive drafting strategies?

**Limitations:**

See weaknesses and questions.

**Strengths And Weaknesses:**

## Strengths

1. Industrial Impact for SOTA Methods: EAGLE-3 is currently the SOTA speculative decoding method, and SpecForge provides a robust, production-ready environment by solving the efficiency bottlenecks of existing frameworks.

2. Memory Efficiency & Long-Context Support: The framework significantly reduces memory overhead (up to 93.5% for attention) during data generation and TTT through custom Triton kernels and Sparse Tree Attention. This allows for training with long-context data, helping to alleviate the acceptance rate degradation in long-text scenarios.

3. Comprehensive Experimental Evaluation: The paper is well-structured and clearly presented. The authors conduct extensive experiments across multiple model families (Llama, Qwen, Kimi) and diverse benchmarks (math, code, general tasks), providing strong empirical evidence for their claims.

4. Practical Ecosystem Contribution: The release of SpecBundle and the distillation of actionable training recipes (e.g., data regeneration impacts) offer immediate practical value for real-world deployment and future research.

## Weaknesses

1. The evaluation is relatively narrow in terms of both task diversity (primarily math and coding) and hardware platforms (primarily H200). More comprehensive benchmarks across different hardware generations and task types would further validate the framework's robustness.

2. Currently, the framework and its optimized kernels are highly tailored to EAGLE-3. The lack of support for other mainstream speculative paradigms or emerging variants, such as Medusa or DFlash, limits its current utility as a unified framework.

[1] Cai T, Li Y, Geng Z, et al. Medusa: Simple llm inference acceleration framework with multiple decoding heads[J]. arXiv preprint arXiv:2401.10774, 2024.

[2] Chen J, Liang Y, Liu Z. DFlash: Block Diffusion for Flash Speculative Decoding[J]. arXiv preprint arXiv:2602.06036, 2026.

---

> ### Author Rebuttal · Authors · 2026-03-31
>
> We thank the reviewers for your constructive feedback and insightful comments. We wish to address your concerns below.
>
> ## Benchmark Diversity
>
> We thank the reviewer for emphasizing benchmark diversity. As noted in Section 6.1 (Line 315), our evaluation covers seven benchmarks across diverse domains, including conversations (MTBench), science (GPQA), mathematics (Math500, GSM8K), coding (LiveCodeBench, HumanEval), and general knowledge (FinanceQA). Due to space constraints, we present detailed coding and math results in the main paper, with the remaining benchmarks in Appendix B. We will clarify this organization in the revision for better visibility.
>
> ## GPU Diversity
>
> We thank the reviewer for pointing out the lack of GPU diversity in our evaluation. This limitation was primarily due to restricted access to different hardware platforms during the initial submission. To address this concern, we have conducted additional experiments on NVIDIA H100 GPU, evaluating Llama-3.1-8B and Qwen3-30B-A3B. The results demonstrate consistent performance trends across GPU architectures. We will include these results in the revision of the paper.
>
> ### 1. Inference Performance on H100
>
> We demonstrate the throughput and acceptance gains under the same EAGLE-3 inference setting (i.e. number of drafting steps = 5, top-k = 1, number of draft tokens = 6, concurrent requests = 8). **The result is given in the format "output-throughput (acceptance length)".**
>
> - Llama3.1-8B
>
> | Setting  | MTBench  | Math500 | GSM8K  | HumanEval | LiveCodeBench  | GPQA | FinanceQA  | FinanceQA |
> |-|-|-|-|-|-|-|-|-|
> | No speculative Decoding | 1044.03 (1)    | 1099.16 (1)    | 934.91 (1)     | 1115.34 (1)    | 1068.58 (1)    | 1082.90 (1)    | 980.53 (1)     | 1083.60 (1)    |
> | Baseline  | 2597.87 (3.57) | 2547.05 (3.22) | 1109.09 (1.77) | 2898.55 (3.76) | 2385.42 (3.11) | 2687.38 (3.39) | 1229.30 (1.78) | 1635.20 (1.85) |
> | Ours| 2513.11 (3.54) | 3621.48 (4.63) | 1490.09 (2.86) | 3299.57 (4.33) | 2966.73 (3.92) | 3138.28 (3.95) | 1433.58 (2.08) | 1632.41 (1.83) |
>
> - Qwen3-30B-A3B
>
> | Setting | MTBench | Math500 | GSM8K | HumanEval | LiveCodeBench | GPQA | FinanceQA | FinanceQA |
> |-|-|-|-|-|-|-|-|-|
> | No Speculative Decoding | 1260.86 (1)     | 1391.19 (1)    | 1085.15 (1)    | 1307.86 (1)   | 1406.97 (1)    | 1348.36 (1)    | 1289.87 (1) | 1083.60 (1)    |
> | Ours | 2107.02 ( 2.83) | 3455.68 (4.50) | 1580.82 (3.01) | 2990.13 (4.38) | 3211.14 (4.02) | 2344.58 (3.00) | 1790.80 (2.25) | 1635.20 (1.85) |
> | Ours | 2513.11 (3.54)  | 3621.48 (4.63) | 1490.09 (2.86) | 3299.57 (4.33) | 2966.73 (3.92) | 3138.28 (3.95) | 1433.58 (2.08) | 1632.41 (1.83) |
>
> ### 2. Training on H100
>
> The batch size is set so to avoid OOM. Our system can still achieve 2x speedup on H100. The speedup is lower on H100 compared to H200 because H100 is smaller memory space, limiting the training batch size per GPU, but this can be resolved if we disaggregate the draft and target models on independent GPUs.
>
> - Llama3.1-8B
>
> | Setting | Batch Size Per GPU | Avg step time/s | Throughput per GPU (tokens/s |
> | - | - | - | - |
> | Baseline ZeRO 2 | 1 | 0.53 | 7728.30 |
> | Ours (TP=2)    | 2 | 0.87 | 9416.09  |
>
> - Qwen3-30B-A3B
>
> | Setting | Batch Size Per GPU | Avg step time/s | Throughput per GPU (tokens/s |
> | - |-|-|-|
> | Baseline ZeRO 3 | 1      | 0.61            | 6714.75                      |
> | Ours (TP=4)     | 2                  | 0.31            | 13212.91                     |
>
> ## Autoregressive draft model vs native MTP
>
> We thank the reviewer for this insightful question on comparisons with native multi-token prediction (MTP). We believe that, with sufficient training data, the autoregressive approach in EAGLE-3 can match or surpass native MTP in acceptance rate. To support this, we evaluated our method on Qwen3-Next, which includes a native MTP module. Despite using a much smaller dataset (PerfectBlend) than Qwen3-Next’s large-scale training data, our approach achieves comparable performance, suggesting that scaling data could further close or exceed the gap. Moreover, many recent open-source models (e.g., Kimi K2.5, MiniMax M2.5) do not provide native MTP, motivating a general and flexible framework like SpecForge that enables efficient speculative decoding without relying on MTP support.
>
> ## Support for Other Methods
>
> Our system is designed as a general framework for target–draft decoupled training and is not restricted to a single algorithm. While prior methods such as Medusa are less competitive compared to more recent approaches (e.g., EAGLE-3), our framework is extensible and can support different speculative decoding algorithms. To demonstrate this extensibility, we have implemented DFlash within SpecForge, leveraging its publicly available training pipeline. This showcases that our system can accommodate diverse methods with minimal modification. We will include the codebase in the supplementary materials for further validation.

---

> > ### Author Rebuttal · Reviewer_BSLo · 2026-04-03
> >
> > Thank you for the detailed response, and I am happy to maintain my supportive score.

---

### Official Review · Reviewer_bJGB · 2026-03-20

**Soundness:** 1
**Presentation:** 2
**Significance:** 3
**Originality:** 1
**Overall Recommendation:** 2
**Confidence:** 2

**Summary:**

SpecForge is a framework for speculative decoding draft models.
The authors have identified that EAGLE-3's limitation is the coupling of a target model from the trainable draft model.
They decoupled them to allow for some degree of parallelism strategies to improve performance.
Additionally, they apparetly utilize sparse tree attention (FlexAttention , masking), which they've implemented using a custom compute kernel.
Most notably, they report up significant inference speedups.

**Compliance With Llm Reviewing Policy:**

Affirmed.

**Ethical Review Concerns:**

Sorry if this is the  wrong way to flag potential issues:

The related work section contains a reference to `Fasteagle` a concurrent submission.
Not only can  this weaken the double-blind process of ICML, but it is literally impossible for me to validate the claims made by the authors or judge a sufficient level of innovation.

**Ethics Expertise Needed:**

["Research Integrity Issues (e.g., plagiarism)"]

**Key Questions For Authors:**

- The Qwen3 result is a major and quite impressive outlier. Being a MoE, shouldn't the cost be much reduced by the sparse activation pattern? Can the authors expand on the explanation?
- Are your improvements over existing checkpoints statistically reliable?
- Can you provide tradeoff plots that depict the speedup relative to varying draft model sizes, draft model acceptance rates, or draft model losses ('accuracies')? That is, can we have a quatities vs. speedup plot to find a sweet splot?

**Limitations:**

- The paper mainly targets the speculative decoding community.
- Relevant concepts, such as EAGLE-3 are less well explained  for people not familiar with the subject.
- It is unclear whether the reported speedups depend on the GPU architecture (i.e., because of sufficient memory bandwidth), of if they are generalizable to other architectures.
- The structural limitations of this approach not well discussed.
- The paper reports speedups based on distinct sources of potential performance gains without a clear attribution. A more extensive ablation study would significantly improve the experiments.
- While some results are likely soemwhat stochastic (Tbl 1, 2), and scores in the bar plots are partially close, tHe paper does not report confidence intervales, making it hard to see the potential significance.
- The paper does not report draft model size, draft model acceptance rate, or draft model loss as a function of inference speedup across the evaluated configurations. From my understanding, these partually determine speedup in Eq. 1 and are highly relevant to be included.
- The related work section is thin
- THe related work section contains a reference to `Fasteagle` a concurrent submission (!!). Not only can  this weaken the double-blind process of ICML, but it is literally impossible for me to validate the claims made by the authors or judge a sufficient level of innovation.
- Finding a the right hyperparameters for TTT might be compilicated and introduce additional costs.
- Competitors focus on EAGLE-like methods, but the paper largely fails to compare with other alternatives. Please elaborate why your limitated evaluation makes sense is enough to be rigorous.

**Strengths And Weaknesses:**

- The speedups are seemingly impressive
- The target–draft decoupling architecture is well motivated
- The paper provides sufficient explanations to reproduce the sparse tree attention kernel code at least in a naive way.

---

> ### Author Rebuttal · Authors · 2026-03-31
>
> We thank the reviewer for your constructive feedback and insightful comments. We wish to address your concerns below.
>
> ## Explanation on MoE Performance
> In Figure 2, our system with the SGLang backend significantly outperforms other backends on Qwen3 MoE models. To understand this gap, we profiled both Hugging Face (https://ibb.co/N2xjbQKV) and SGLang (https://ibb.co/SwTLctwk) backends. The results show that the Hugging Face backend suffers from sparse kernel launches, which severely degrades prefill performance during hidden state generation. This highlights the importance of integrating a production-grade inference engine. By leveraging high-performance kernels and CUDA Graphs, our system significantly improves prefill efficiency and removes inference bottlenecks in the training pipeline.
>
> ## Statistical Relibility
> We confirm our results are statistically reliable. Each benchmark was run 5 times, reporting average throughput with warmup for stable profiling. We rerun the inference and training experiments on Llama3-8B and Qwen3-30B-A3B on H200 GPUs and obtained similar results as reported in the paper.  We have also reported some results on H100 GPUs,  **please see the section "GPU Diversity" in my rebuttal to reviewer BSLo for H100 performance**.
>
> ## Report of Draft Model Size, Acceptance Rate and Trade-off Plot
> We provide a plot (https://ibb.co/XrrRfHyj) showing the relationship between draft model size, acceptance length, and speedup (using GSM8K). While larger draft models generally yield higher acceptance length, this does not always translate to higher speedup, which depends on the target model architecture. In particular, dense models benefit more, whereas for MoE models, verifying multiple draft tokens requires loading the union of activated experts, increasing memory cost and reducing speedup.
>
> We did not report acceptance length in the paper as we reported the best performance among various EAGLE-3 configurations for both baseline and our system; direct comparison is not always meaningful as acceptance length increases with more draft tokens. Nevertheless, we agree this metric is important to demonstrate performance preservation.
>
> ## Research Topic
> We believe speculative decoding is a major and broad topic for LLMs due to its strong, lossless acceleration. We apologize for not fully explaining key concepts like EAGLE-3 due to page limits and will include a detailed appendix with figures in the revision.In brief, earlier methods relied on target-model hidden states, causing a training–inference mismatch. EAGLE-3 addresses this by using its own hidden states to generate multiple tokens, reducing training-inference mismatch.
>
> ## Ablation Studies
> May we seek your clarification on the type of ablation study that is required?
>
> ## Related Work and Limitations
> We would add more discussion of these sections in our next revision. One limitation of our method is that it only works with hidden-states-based speculative decoding methods, however these methods are currently the mainstream methods/
>
> ## GPU Architecture
> The exact speedup varies with GPU architecture, but speculative decoding and our training system consistently improves performance across modern GPUs. Moreover, memory bandwidth does not scale proportionally to compute in modern GPUs. We validate this on H100 and H200 GPUs (see “Statistical Reliability”).
>
> ## Concurrent Submission
> We have fully adhered to ICML’s concurrent submission guidelines by including our concurrent work, FastEAGLE, in the supplementary material and citing it in the Related Work section. Our submission satisfies all requirements, including treating it as prior work and providing an anonymized PDF. Importantly, the two works differ in scope: FastEAGLE addresses storage challenges in offline training, while our work focuses on scalability and efficiency in online training. The contributions are therefore complementary rather than overlapping. We believe this fully complies with ICML policies and raises no ethical concerns.
>
> Ref:
> https://icml.cc/Conferences/2026/CallForPapers
> https://icml.cc/Conferences/2026/PeerReviewEthics#concurrent
>
> ## TTT parameter search
> TTT hyperparameter search can be conducted on a small subset of data. In practice, although draft models are trained on millions of samples (e.g., Perfect Blend with 1.4M samples), we find that using ~50k samples is sufficient to identify suitable hyperparameters for the full training run.
>
> ## Comparison with other methods
> Our framework targets scalability and efficiency rather than new speculative decoding methods. We use EAGLE-3 as the main benchmark due to its state-of-the-art performance and broad adoption, while earlier methods (e.g., Medusa, EAGLE-1/2) are less competitive. The framework is extensible and supports new approaches; for instance, we integrated DFlash-based training. As DFlash was released after submission and is also under ICML review, we include it only in the supplementary material.

---

> > ### Author Rebuttal · Reviewer_bJGB · 2026-04-03
> >
> > I acknowledge your rebuttal and understand your point of view, but it doesn't change my initial assessment and my primary concerns. I am more comfortable with keeping my evaluation largely as is.
> >
> > Please do note that I highlighted the concurrent submission as a reason for the reviewer's inability to judge the correctness of your claims and novelty of your work due to the lack of access to your related work.

---

> > > ### Author Response · Authors · 2026-04-04
> > >
> > > We thank the reviewer for the time and effort for acknowledging our rebuttal. We would like to use this final response to (1) address the point regarding concurrent submission, (2) seek clarification on the remaining concerns, and (3) briefly summarize how we have responded to the main technical questions.
> > >
> > > ## Clarification regarding concurrent submission
> > >
> > > The reviewer mentions that the concurrent submission limited your ability to assess correctness and novelty due to **“lack of access.”** We would like to clarify:
> > > -  We stated in our previous response that **we included an anonymized PDF of the concurrent work in the supplementary material**, as part of our efforts to fully comply with ICML’s concurrent submission policy.
> > > -  The concurrent work was **available** to reviewers since  paper submission as an anonymous PDF, and **can be downloaded and examined** to verify our claims and results.
> > > -  Therefore, our concurrent work **does not** hinder the reviewer's ability to assess our work due to the inability to access our related work.
> > >
> > > We are concerned that this point may have led to an unintended misunderstanding that affected the assessment of our current paper.
> > >
> > > ## Request for clarification on remaining concerns
> > >
> > > We note that the reviewer indicates that the rebuttal is "partially resolved or unresolved". However, the acknowledgement does not specify:
> > > -  which concerns have been resolved,
> > > -  which concerns remain unresolved, and
> > > -  why the rebuttal was insufficient to address them.
> > >
> > > Given the limited interaction rounds in the rebuttal process, this lack of specificity makes it difficult for us to meaningfully address the reviewer’s remaining concerns. Even though we should have no more chance to reply the reviewer's comment, we respectfully request clarification on the exact issues that are considered unresolved.
> > >
> > > ## Summary of responses to key technical concerns
> > >
> > > To ensure clarity, we briefly summarize how we have addressed the main questions raised:
> > >
> > > - MoE Performance
> > >
> > > The observed performance gap is not primarily due to MoE sparsity, but due to backend efficiency. Through profiling of Hugging Face and SGLang backends, we show that Hugging Face suffers from sparse kernel launches, leading to significant prefill inefficiency during hidden state generation. By integrating a production-grade inference engine (SGLang) with optimized kernels and CUDA Graphs, our system eliminates this bottleneck and achieves substantially higher throughput.
> > >
> > > - Statistical Reliability and GPU Diversity.
> > >
> > >  All reported results are averaged over 5 runs with controlled settings and warmup to ensure stable profiling. We further validated our results by re-running both training and inference experiments on Llama3-8B and Qwen3-30B-A3B across H200 and H100 GPUs, observing consistent speedups. Additional H100 results are provided in our rebuttal, confirming robustness across hardware platforms.
> > >
> > >
> > > - Draft Model Size, Acceptance Rate, and Trade-offs.
> > >
> > > We explained the source of speedup in the experiments sections, such as the draft-target decoupling over unified parallelism strategy and the performance comparison of different backends for hidden states generation. As for the model performance in terms of speedup, we also provided a trade-off plot in our initial response with explanation.
> > >
> > > - Scope and Research Positioning.
> > >
> > >  Our work focuses on speculative decoding, which is a central and widely adopted paradigm for lossless LLM acceleration. We acknowledge that key concepts may not have been fully explained due to space constraints and will expand these explanations (with figures) in the revision.
> > > Generalization Across GPU Architectures.
> > >  While absolute speedups vary with hardware characteristics, our approach consistently improves performance across modern GPUs. We empirically validate this on both H100 and H200 GPUs.
> > >
> > > - TTT Hyperparameter Search.
> > >
> > >  TTT hyperparameters can be efficiently tuned on a small subset of data. In practice, although full training may use millions of samples, we find that ~50k samples are sufficient to identify suitable configurations.
> > >
> > > - Comparison with Other Methods.
> > >
> > >  Our contribution is a training framework rather than a new speculative decoding algorithm. We focus on EAGLE-3 as it is the current state-of-the-art and widely adopted baseline. The framework is extensible to other methods; for example, we have integrated DFlash-based training (included in supplementary material due to timing and concurrent submission considerations).
> > >
> > > We sincerely appreciate the reviewer’s feedback. However, we respectfully note that:
> > > -  The remaining concerns are not clearly specified, and
> > > -  key points (e.g., concurrent submission access and statistical reliability) may have been based on misunderstandings that we have clarified in our initial response.
> > >
> > > We hope this final response helps provide additional clarity for a fair assessment of our work.

---

### Official Review · Reviewer_tCLT · 2026-03-21

**Soundness:** 2
**Presentation:** 3
**Significance:** 3
**Originality:** 2
**Overall Recommendation:** 4
**Confidence:** 4

**Summary:**

This paper proposes SpecForge, a system framework for training draft models for speculative decoding (e.g., EAGLE-3). The key techniques include target–draft decoupling, hybrid parallelism, optimized training kernels, and tight integration with production-grade inference engines. The work also introduces SpecBundle, a suite of production-grade draft models that significantly accelerate inference for mainstream open-source LLMs. Experimental results show that both training throughput and inference speed are significantly improved.

**Compliance With Llm Reviewing Policy:**

Affirmed.

**Key Questions For Authors:**

See the weakness section

**Limitations:**

See the weakness section

**Strengths And Weaknesses:**

Strength:
1. The paper addresses a practical and important problem: scalable training of speculative decoding draft models.

2. The paper demonstrates strong engineering and system design. In particular, the target–draft decoupling and hybrid parallelism are well-motivated and effectively address the mismatch between inference-heavy and training-heavy workloads. The proposed sparse tree attention and custom Triton kernels significantly reduce memory usage and improve training efficiency.

3. The paper includes thorough empirical evaluation, covering both training throughput (Table 1) and downstream inference performance (Table 2). The release of SpecBundle and training recipes may be useful for practitioners.

Weakness:
1. The technical novelty is somewhat limited, as core components like decoupled execution, hybrid parallelism, and sparse attention are well-established system optimizations. The contribution lies primarily in the engineering integration for speculative decoding rather than a fundamentally new design.

2. It is unclear whether the proposed optimizations affect model performance. In Table 2, the authors evaluate performance using draft models trained on Open-PerfectBlend with regenerated responses, while existing baselines are trained on ShareGPT. The author’s dataset for training is much larger and more aligned with math and coding tasks compared with baseline, which will lead to better performance regardless of the system design. This makes it difficult to isolate the effect of the proposed training framework.

3. Some insights are not sufficiently justified. For example, in Section 7.1, the reported ~5.3% throughput improvement from data regeneration appears consistent with the already observed speedup increase from EAGLE (e.g., 2.78 => 2.88), suggesting this insight largely restates empirical results rather than providing new analysis.

---

> ### Author Rebuttal · Authors · 2026-03-31
>
> We thank the reviewer for your constructive feedback and insightful comments. We wish to address your concerns below.
>
> ## Novelty
>
> While we agree that components such as hybrid parallelism and sparse attention are individually known, our contribution is not a direct combination, but a new system design tailored to speculative decoding training, which has a fundamentally different workload from standard training or inference. Specifically, speculative decoding introduces a dual-model asymmetry (draft vs. target) and TTT-induced autoregressive computation, for which existing frameworks are suboptimal.
>
> Our key novelty is target–draft decoupling as a first-class abstraction, enabling heterogeneous parallelism and, importantly, the integration of a production-grade inference engine (SGLang) into the training loop, which prior work does not support .
> Moreover, our optimizations (e.g., TTT-specific sparse attention and memory-efficient training kernels) are co-designed with this execution model, rather than reused directly. The resulting up to 9.9× speedup suggests this is not merely engineering integration but addresses previously unrecognized system bottlenecks.
>
> ## Impact of SpecForge on Model Performance
>
> We thank the reviewer for raising this important point. We emphasize that our system optimizations are performance-preserving, and we have verified this through controlled experiments.
>
> First, for kernel-level optimizations, we implement unit tests comparing our kernels with torch-native standard implementations under both BF16 and FP16 data formats. The numerical differences remain within standard tolerance and do not affect downstream performance.
>
> Second, to isolate the effect of the training framework, we train LLaMA-3-8B using both SpecForge and the original EAGLE-3 codebase on the same ShareGPT dataset with identical hyperparameters (i.e., batch size per GPU 4, learning rate 1e-4, 2 epochs, cosine annealing warmup LR schedule).
>
> We have placed a loss curve figure in the anonymous link https://ibb.co/3mjnWFj8 to show that the baseline and our system show closely matching convergence. The orange curve represents the baseline and pink curve represents our system. In addition, we have also evaluated the models trained with the baseline and our system respectively on all benchmarks shown in the paper, under the same EAGLE-3 settings (number of drafting steps 3, top-k 1, number of draft tokens 4).  The resulting models achieve comparable performance across all evaluated benchmarks, indicating that our system design does not alter model quality.
>
> **The result is given in the format "output-throughput (acceptance length)".**
>
> | Model     | Setting                      | MTBench        | Math500        | GSM8K          | HumanEval      | LiveCodeBench  | GPQA           | FinanceQA      |
> |-----------|------------------------------|----------------|----------------|----------------|----------------|----------------|----------------|----------------|
> | Llama3-8B | Without Speculative Decoding | 1283.62 (1)    | 1343.99 (1)    | 1231.00 (1)    | 1348.83 (1)    | 1303.98 (1)    | 1318.24 (1)    | 1083.60 (1)    |
> | Llama3-8B | Baseline                     | 2335.49 (2.49) | 2905.37 (2.74) | 1998.38 (2.28) | 2867.89 (2.81) | 2686.98 (2.71) | 2972.41 (2.86) | 1635.20 (1.85) |
> | Llama3-8B | SpecForge                    | 2334.64 (2.49) | 2916.42 (2.75) | 2012.74 (2.32) | 2898.16 (2.84) | 2685.74 (2.71) | 3002.29 (2.86) | 1632.41 (1.83) |
>
> We will revise the paper to explicitly highlight these controlled experiments and clarify that our system preserves model performance through strict numerical validation.
>
> ## Insights Analysis
>
> Our intention in Section 7.1 is not merely to restate empirical improvements, but to challenge a key assumption made in prior work. The original EAGLE-3 paper claims that speculative decoding performance is largely insensitive to the training data distribution and does not require response regeneration.
>
> Our results show that data regeneration consistently yields a non-negligible throughput gain (~5.3%), even when acceptance rates appear only marginally improved (e.g., 2.78 → 2.88). This indicates that small changes in acceptance behavior can translate into amplified system-level speedups, due to the nonlinear relationship between acceptance rate and end-to-end throughput.
>
> We agree this insight was under-explained and will revise the paper to better articulate why this effect arises and its practical implications for large-scale deployment.

---

> > ### Author Rebuttal · Reviewer_tCLT · 2026-04-07
> >
> > i've read the rebuttal and think it is clear. I maintain my positive assessment of this work!

---

### Decision · Program_Chairs · 2026-04-30

**Decision:**

Accept (regular)

**Comment:**

Comment: The review from reviewer bJGB is a low-quality review, and hence is discarded when writing this meta-review.

The paper proposes SpecForge, an open-source training framework for speculative decoding drafters, with extensive optimization for EAGLE-3 drafters. With carefully optimized training framework, the paper improves EAGLE-3 training baseline by up to 9.9x on Qwen3 model. The paper also releases SpecBundle, a collection of production-level EAGLE-3 drafters for mainstream open-sourced models.

The reviewers all agreed that the paper makes strong engineering and system-level optimizations, including target–draft decoupling, hybrid parallelism, and optimized training kernels. The reviewers also praised the release of production-level drafters for major open-sourced models. The main concerns from the reviewers were centered around the novelty of the work, and the breadth of the empirical evaluations since the current implementation mainly focuses on EAGLE-3 models, and limited task and hardware settings.

During the rebuttal, a few of the above concerns are addressed, and all reviewers maintained their positive assessment of the paper. However, while the paper makes nontrivial system-level contributions, the concerns about the algorithmic novelty and the lack of evaluation on drafter configurations other than EAGLE-3 still stand. Overall I agree with the reviewers concerns and I view the contributions of this work mainly as strong engineering and system-level optimizations.

This being said, I do see the outcome of the work being valuable to the community for the following reasons: (1) EAGLE-3 is arguably the SOTA drafting technique at the moment, and hence this limitation should not be a major concern; (2) While ICML community traditionally values algorithmic contributions more, I do see system-level contributions are becoming increasingly important given the current scale of LLM models. This is particularly important for the LLM efficiency field. This work stands out in this regard.

Overall, I would recommend weak acceptance for the paper. I encourage authors to incorporate the reviewers' comments to revise the draft. Demonstrating the generality of the proposed optimizations beyond EAGLE-3 would make the contribution stronger and more general.